

# Retrieval simulations of a spaceborne differential absorption radar near the 380 GHz water vapor line

Luis F. Millán[1], Matthew D. Lebsock[1], and Marcin J. Kurowski[1]

[1]Jet Propulsion Laboratory, California Institute of Technology, Pasadena, CA, USA

**Correspondence:** Luis Millán (lmillan@jpl.nasa.gov)

**Abstract.** Differential Absorption Radar (DAR) is an emerging technique for high resolution humidity profiling inside clouds and precipitation. This study evaluates the potential of using a spaceborne DAR operating near the 380 GHz water vapor absorption line to profile water vapor in the mid and upper troposphere, particularly inside deep convective systems. To quantify the expected precision and accuracy of DAR and to define optimal channel selection, we modeled radar reflectivities from
large-eddy simulation fields and then implemented retrievals using the simulated observations.

End-to-end retrieval simulations across the 350–380 GHz range were used to identify optimal radar frequency triplets, minimizing precision and biases, at each altitude. Each optimum triplet included the most transparent frequency available, with the other two radar tones varying with altitude. At higher altitudes the optimization identifies frequencies close to the line center and the optimum frequencies move progressively away from the line at lower altitudes. Results show that single-pixel
(horizontal resolution $\simeq 400$ m and vertical resolution $= 200$ m) precision generally exceeds 100% with biases typically below 10%. Precision can be enhanced by averaging along-track. For instance, by optimizing the triplet selection, a precision of 0.01 $\text{gm}^{-3}$ can be achieved by averaging over 50 km in anvil outflows with extensive cloud coverage. We note that the improvement may be less than expected in scenarios where cloud coverage is limited since the DAR technique only works in cloudy volumes.

Lastly, we use real world clouds observed by CloudSat to quantify global yield. Most radar tones examined here achieve a
global sampling yield of over 95% at their target altitude. When developing a DAR instrument, selecting the appropriate triplet is essential, taking into account the target altitude and cloud types intended for observation.

*Copyright statement.* ©2025. California Institute of Technology. Government sponsorship acknowledged.

## 1 Introduction

Knowledge of the vertical profile of water vapor is crucial for understanding cloud and precipitation microphysics, atmospheric
radiative transfer, land–atmosphere interactions, and for improving weather forecasts and climate change projections. However, observing water vapor in and around convective storms using infrared and microwave sounders is severely limited by the confounding effects of clouds and precipitation on observed brightness temperatures (e.g., Greenwald and Christopher, 2002; Fetzer et al., 2006). Further, while radiooccultation (e.g., Kursinski et al., 1997; Wang et al., 2024) and microwave limb





emission measurements (e.g., Read et al., 2007; Eriksson et al., 2014) can provide relatively high vertical resolution water vapor
profiles, their measurement geometry averages over horizontal distances exceeding hundreds of kilometers. This observational
gap hampers our ability to constrain the water budget of convective systems, thereby increasing uncertainty in estimates of
convective mass transport, associated cloud microphysical processes, and overall water vapor feedback to climate. Differential
Absorption Radar (DAR) measurements (e.g., Lebsock et al., 2015; Millán et al., 2016; Roy et al., 2018; Battaglia and Kollias,
2019) could help address this gap by profiling mid and upper tropospheric water vapor within convective cloud systems.

The DAR concept is analogous to the differential absorption lidar (DIAL) technique (e.g., Browell et al., 1983; Wulfmeyer
and Bösenberg, 1998; Behrendt et al., 2009; Carroll et al., 2022). In essence, these techniques estimate the gas absorption by
comparing backscatter signals at frequencies "on" and "off" an absorption line and at different ranges from either a laser or
radar pulse. The efficacy of the DAR technique to estimate water vapor profiles under cloudy conditions has been demonstrated
from the ground (Cooper et al., 2011; Roy et al., 2018) and from aircraft platforms (Roy et al., 2022; Millán et al., 2024) using
radar tones around the 183 GHz water vapor absorption line. However, radar transmissions near the 183 GHz line center are
internationally prohibited (NTIA, 2023), restricting the existing DAR operations to the 158-174.8 GHz range and limiting their
sensitivity to lower tropospheric clouds, where water vapor abundance is relatively large.

Unregulated water vapor lines at 325 and 380 GHz (see Figure 1) could extend the DAR technique to mid and upper tro-
pospheric clouds, including those associated with convection and stratiform anvils. These lines are strongly absorbing, max-
imizing DAR sensitivity in these regions where vapor abundance (and thus absorption) is low. In this study, we assess the
effectiveness of the DAR technique in retrieving water vapor through the analysis of end-to-end retrievals using radar tones
near the 380 GHz line. Following Roy et al. (2021), we assume an instrument with a 2 meter diameter antenna, a high-power,
high-duty-cycle transmitter with peak output power of 100 W, and a long-duration pulse with linear frequency modulation and
pulse compression with very large time-bandwidth (similar to the RainCube mission radar, Peral et al., 2019). Such transmitter
performance is consistent with a vacuum-electronics-based, traveling-wave tube amplifier currently under development (e.g.,
Bian et al., 2021).

## 2 Methodology

### 2.1 Large-Eddy simulations

The simulated radar reflectivities analyzed in this study were calculated for atmospheric states supplied by means of large-
eddy simulation (LES). LES is a well-established method for producing realistic high-resolution three-dimensional fields of
temperature, moisture, clouds, precipitation, and winds (e.g., Stevens and Lenschow, 2001; Stoll et al., 2020; Kurowski et al.,
2023). In particular, we used the Weather Research and Forecasting (WRF)-LES model (Skamarock et al., 2008) with the
initial and boundary conditions from the GARP Atlantic Tropical Experiment (GATE, Betts, 1974), which represents a deep
convective system developing over ocean away from mesoscale disturbances (Roy et al., 2021; Kurowski et al., 2024). This
setup enables the evaluation of the DAR technique's performance, specifically for upper tropospheric clouds forming around
convective clusters. Since the freezing level is located at around 4.5 km, most of high-level clouds contain ice.



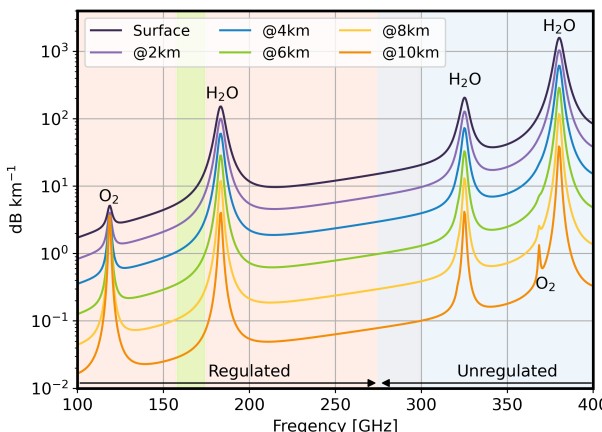

**Figure 1.** Water vapor absorption coefficient at different heights for the average pressure, temperature, and water values for the simulations used in this study (see section 2.2). Red shading indicates internationally regulated regions, light blue shading represents unregulated regions, and green shading shows the existing DAR instrument frequencies for which a special transmit permit over the US was granted.

The LES model applies 100-m horizontal grid spacing and a stretched vertical grid spacing ranging from around 60 m near the surface up to a few hundred meters in the upper troposphere. The domain size is $150 \times 150 \, \text{km}^2$. Atmospheric dynamics is driven by nearly constant surface latent and sensible heat fluxes, with the ocean surface temperature of around 300 K. Verti-
cal transport is dominated by organized deep convection forming due to interactions between updrafts, clouds, precipitation, downdrafts, and cold pools (cf. Kurowski et al., 2024). We use one-moment bulk microphysics to represent six classes of water: vapor, liquid and ice clouds, rain, snow, and graupel. For simplicity, a weak long-wave radiative cooling is prescribed and the diurnal cycle is neglected.

Lastly, the three-dimensional LES outputs are combined with upper-atmospheric fields from Modern-Era Retrospective
Analysis for Research and Applications (MERRA-2) reanalysis (Gelaro et al., 2017) to produce full atmospheric columns to be used as input for the radar forward model. More details about the LES-based retrieval framework can be found in Kurowski et al. (2023) and references therein.

## 2.2 Radar forward model

Radar returns were simulated using the same radar forward model as discussed in Roy et al. (2021). In short, radar reflectivities
were estimated using the time-dependent two-stream approximation (Hogan and Battaglia, 2008), assuming spherical particles for cloud, rain and graupel hydrometeors, hexagonal columns for ice, and dendrites for snow (Leinonen and Szyrmer, 2015). Gas absorption was modeled using the clear-sky forward model for the Microwave Limb Sounder (Waters et al., 2006), as described in Read et al. (2004), which incorporates both line-by-line and continuum absorption contributions. Lastly, scattering





**Table 1.** Radar forward model specifics

| Parameter | Reference |
| --- | --- |
| Cloud particle size distribution | Miles et al. (2000) |
| Rain particle size distribution | Abel and Boutle (2012) |
| Ice particle size distribution | Cox (1988) |
| Snow particle size distribution | Wood (2011) |
| Graupel particle size distribution | Field et al. (2019) |
| Water dielectric properties | Liebe et al. (1991) |
| Ice dielectric properties | Warren and Brandt (2008) |
| Mie scattering | Bohren and Huffman (1998) |
| T-matrix calculations | Leinonen (2014) |
| Radiation propagation | Hogan and Battaglia (2008) |
| | Cox and Munk (1954) |
| Surface reflection | Meissner and Wentz (2004) |
| | Roy et al. (2022) |

from the ocean surface was computed using a quasi-specular scattering model (e.g., Valenzuela, 1978). Additional details, such

as the dielectric constants and particle size distributions employed, can be found in Table 1.

These simulated radar returns leveraged the fine spatial resolution of LES simulations to simulate non-uniform beam-filling effects both within individual range bins and across the horizontal footprint. The average backscatter of the antenna footprint was computed by performing a discrete summation over the fine resolution of the LES. The radar horizontal footprint was defined by the 3dB full width of the two-way beam antenna pattern and by the product of the integration time and ground speed.

Note that, for comparing radar-retrieved humidity profiles with those from the LES, we also averaged the model humidity field over the time-averaged two-way antenna pattern.

The relative uncertainty in the reflectivity measurements due to random errors was modeled following the radar speckle model for randomly distributed scatterers (e.g., Papoulis, 1965; Doviak and Zrnić, 1993; Torres, 2001; Roy et al., 2018), accounting for the speckle noise, the Townes noise (i.e., Townes and Geschwind, 1948; Pearson et al., 2008), and the instrument

thermal noise. The instrument parameters that determine the minimum detectable signal and random measurement uncertainty are listed in Table 2.

Figure 2a illustrates a cross-section of the LES-driven radar simulations, showcasing a deep convective cloud present throughout the domain. Panels b and c show the radar reflectivities at the edge of the frequencies explored in this study, that is, 350 and 380 GHz, respectively. These frequencies correspond to the least and most attenuated frequencies considered.

The impact of the water vapor *continuum* can be observed in the 350 GHz cross section, where the radar signal can only penetrate the cloud column up to around 2.5 km. The additional attenuation due to the nearby water vapor line is evident in the 380 GHz cross section, where the radar signals can only penetrate up to around 11 km. These cross sections (panels b and



**Table 2.** Radar system parameters

| Parameter | Value |
| --- | --- |
| Transmit power | 100 W |
| Duty cycle | 25 % |
| Pulse duration | 50 $\mu$s |
| Pulse repetition interval | 200 $\mu$s |
| Range resolution | 50 m |
| Antenna diameter | 2 m |
| Receiver noise figure | 8 dB |
| Orbital altitude | 400 km |
| Along-track integration time | 60 ms |
| Along-track footprint | 400 m |
| Number of pulses per frequency | 92 |
| Minimum detectable reflectivity | $\sim$-43 dBZ |

c) exemplify why the DAR technique at these frequencies (350-380 GHz) may be better suited to study ice clouds. At these frequencies, not only is the vapor attenuation significant, but also the attenuation from liquid is very large.

## 2.3 Water vapor retrieval

The DAR retrieval methodology is fully discussed elsewhere (e.g., Battaglia and Kollias, 2019; Roy et al., 2020). In short, it starts by combining the observed reflectivities to form the observed absorption coefficient, $\gamma$, at two different ranges $r_1$ and $r_2 = r_1 + R$,

$$\gamma(r_1, r_2, f) = \frac{1}{2R} ln \left[ \frac{Z_e(r_1, f)}{Z_e(r_2, f)} \right] \tag{1}$$

where $Z_e(r, f)$ is the measured reflectivity given by $Z_e(r, f) = Z_{eff}(r, f)e^{-2\tau(r,f)}$, with $Z_{eff}(r, f)$ being the effective unattenuated reflectivity for a given target, and $\tau(r, f)$ being the one-way optical depth from the radar to the range $r$ at frequency $f$.

For small R and assuming negligible multiple scattering, the fitting function for $\gamma(r_1, r_2, f)$ has been shown to be (e.g., Battaglia and Kollias, 2019; Roy et al., 2020),

$$\gamma(r_1, r_2, f) = \overline{\rho_v}(r_1, r_2)\kappa_v(f) + \frac{1}{2R} ln \left[ \frac{Z_{eff}(r_1, f)}{Z_{eff}(r_2, f)} \right]$$
$$+ \overline{\beta_{dry}}(r_1, r_2, f) + \overline{\beta_p}(r_1, r_2, f) \tag{2}$$

where $\overline{\rho_v}$ is the water vapor density, $\kappa_v(f)$ is the water vapor mass extinction coefficient, $\overline{\beta_{dry}}(r_1, r_2, f)$ is the dry-air absorption coefficient, $\overline{\beta_p}(r_1, r_2, f)$ is the particulate extinction coefficient, and where the overline indicates taking the average between $r_1$ and $r_2$.




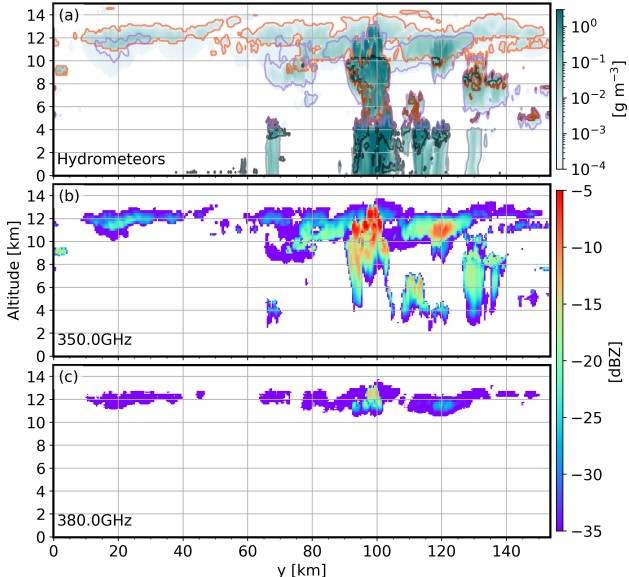

**Figure 2.** Cross sections illustrating the LES-driven simulations: (a) LES total hydrometeor water content (IWC+LWC+rain+snow) . Orange, black, gray, and purple lines, respectively delimit areas where IWC, LWC, rain, and snow were present. (b) Simulated LES-driven radar reflectivity at 350 GHz. (c) Simulated LES-driven radar reflectivity at 380 GHz.

This equation implies that by measuring the observed absorption coefficient, $\gamma(r_1, r_2, f)$, at different frequencies it is possible to fit the terms on the right-hand side of equation. The first term is directly proportional to the water vapour density via the water vapor mass extinction coefficient. The remaining parameters, which vary weakly with frequency, provide information about the relative reflectivity of the two ranges in question: dry air gaseous absorption and particulate extinction, respectively. Note that since $\gamma(r_1, r_2, f)$ is the ratio of reflectivities at two different heights, it is not affected by absolute calibration, making in-cloud humidity estimation unaffected by calibration errors.

Assuming the last 3 parameters in equation 2 are a frequency-independent offsets, the humidity can be estimated directly using measurements of $Z_e$ at two frequencies (e.g., Roy et al., 2018). However, this assumption does not hold true in areas where the particle size distributions of hydrometeors vary greatly, such as near cloud boundaries or in regions experiencing phase changes. In these areas, the range-dependent differential scattering of hydrometeors could be mistakenly attributed to water vapor attenuation, leading to humidity biases in the retrievals (e.g., Roy et al., 2020; Millán et al., 2024).

Battaglia and Kollias (2019) and Roy et al. (2020) demonstrated theoretically that with $Z_e$ measurements at a minimum of three frequencies, it is possible to partially disentangle the differential extinction due to water vapor from the scattering and absorption effects of hydrometeors. The rationale behind this bias mitigation is that the water vapor mass extinction coefficient, $\kappa_v(f)$, may exhibit a pronounced curvature with respect to frequency, while the rest of parameters in equation 2





vary approximately linearly with frequency. This allows the retrieval to distinguish between the two distinct contributions.
Millán et al. (2024) demonstrated the effectiveness of this bias mitigation using real data.

The aim of this study is to explore various frequency combinations to identify triplets that not only reduce such biases but that also minimizes the retrieved humidity precision (i.e., those associated with the reflectivity random errors). To achieve this goal, we conducted end-to-end retrieval simulations assuming a retrieval step, $R$, of 200 m. The retrieval algorithm employed in this study is fully described in Roy et al. (2021).

Figure 3 shows examples of retrieval estimates for different frequency triplets, selected to show how different triplets can sample different regions of the LES cloudy volume (shown in panel a). Triplets close to the line center (e.g., 370, 375, and 380 GHz, panel b) can probe higher altitudes. In contrast, triplets slightly further away from the line center (e.g., 360, 365, and 370 GHz, panel c) can penetrate further toward the surface. Lastly, triplets with moderate water vapor absorption (e.g., 350, 355, and 360 GHz, panel d) can penetrate even deeper towards the surface but sampling of the upper parts of clouds is noisy
and uneven due to the lack of spectral contrast.

These retrievals have an associated precision, bias, and yield. The precision represents the random errors as estimated by the mapping of the random radar *measurement* errors into the retrieved humidity values. The bias represents the difference between the retrieved values and the actual LES values. These biases are associated with frequency-dependent hydrometeor extinction and backscatter (Roy et al., 2021), and non-uniform beam filling. The yield is simply given by,

$$Y = \frac{N_M}{N_T} \tag{3}$$

where $N_M$ is the number of times radar reflectivities were measured at the three frequencies, and $N_T$ is the total number of cloudy volumes at a given height. Essentially, the yield indicates how frequently a particular triplet can effectively sample that level.

Figure 3e-g displays the humidity profiles, precision, and bias, at the location indicated by a black dashed line in Figure 3b-d.
The yield is also implicitly shown by the difference in vertical coverage between the retrieved and the model values. There is an obvious trade-off between precision, bias, and yield, with the 370, 375, 380 GHz triplet offering the best precision and the smallest bias but with limited vertical coverage. Meanwhile, the 350, 355, 360 GHz triplet offers the most comprehensive vertical coverage but with significantly poorer precision and larger biases at high altitudes. These biases are associated with the low water vapor values (i.e., low attenuation) at these heights, which reduce the contrast between the triplets used, and
therefore weaken the signal used to retrieve water vapor.

## 3 Results

We conducted end-to-end retrieval simulations for each frequency triplet in the 350 to 380 GHz region (with a 1 GHz spacing), similar to those shown in Figure 3, encompassing the entire GATE domain. At each height, these retrievals provide estimates of precision, bias, and yield. To maximize the information content, we searched for triplets that minimized both precision and
bias. To achieve this, we normalized the precision for each triplet at every height by the maximum precision found across all





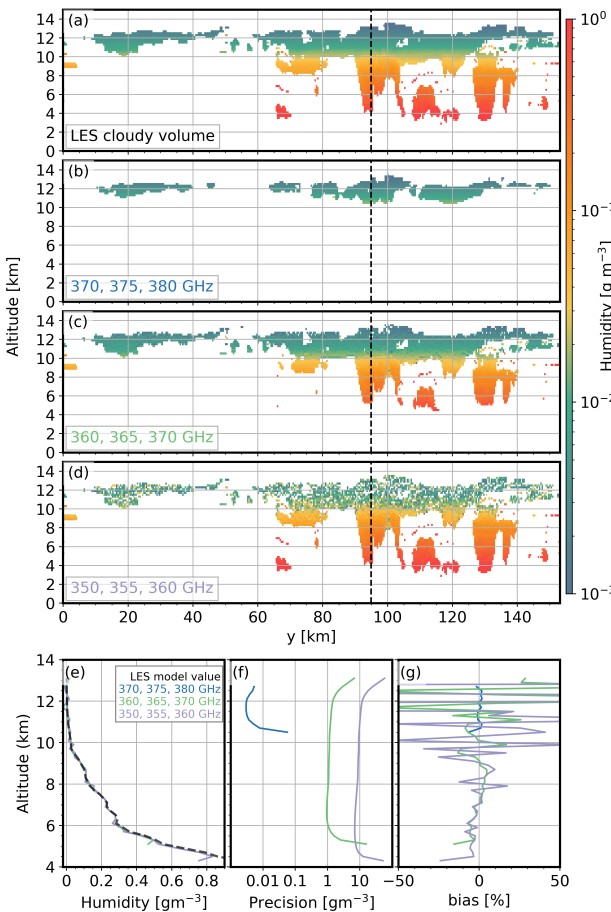

**Figure 3.** Cross sections illustrating the LES humidity simulations subsampled at the cloudy volumes (a) and the DAR retrievals using different frequency triplets: (b) 370, 375, 380 GHz, (c) 360, 365, 370 GHz, and (d) 350, 355, 360 GHz. Examples of DAR retrievals (e), precision (f), and bias (g). These profiles are from the location indicated by the black dashed line in the cross sections.

simulations. That is,

$$\sigma'_{\mathbf{e}} = \frac{\sigma_{\mathbf{e}}}{\max(\sigma_{\mathbf{e}})} \tag{4}$$

where $\sigma_e$ is a vector of the precision estimates at a given height for all the end-to-end retrieval simulations. Similarly, we normalized the bias by,

$$\mathbf{b}' = \frac{\mathbf{b}}{\max(\mathbf{b})} \tag{5}$$

where $\mathbf{b}$ is the corresponding vector of biases at the same height. Then, we simply identified the triplet that produced the retrieval with the minimum value of the sum of $\sigma'_{\mathbf{e}} + \mathbf{b}'$ (i.e., the best combined precision and accuracy) at that level.





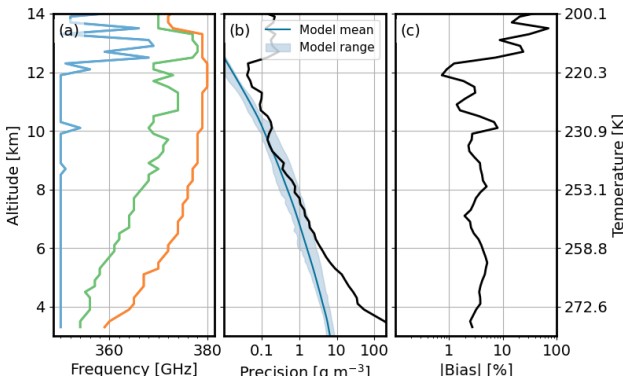

**Figure 4.** (a) Optimal triplet frequencies at different heights for the instrument used here for the GATE LES fields. (b) Precision for such triplets. The mean model values (blue line) and the range of model humidity (light blue shaded areas) are shown for reference. (c) Biases for such triplets. Results are shown for triplets with a yield $> 0.5$ throughout the LES domain.

Figure 4a-c shows the triplets per height that maximize the information content as well as the corresponding precision and biases, enforcing a yield greater than 0.5. At altitudes below 12 km, the optimization algorithm consistently selects, as part

of the triplet, the frequency furthest away from the 380 GHz water vapor line, effectively selecting the edge of the explored range. In other words, regardless of the height analyzed, the selected triplet always includes the most transparent available radar frequency. To the left of the 380 GHz water vapor line (i.e., the spectrum region explored), the closest strong line is another water vapor line centered at 325 GHz. As shown in Figure 1, the most transparent region between these lines is around 340 GHz. Thus, future DAR instruments operating in the 380 GHz region should consider this during their development.

The spacing between the other two frequencies in the triplet ranges from 8 to 10 GHz. At lower altitudes, the algorithm selects frequencies further from the water vapor line center to compensate the increased water vapor absorption. For instance, below 4 km, the two additional frequencies in the triplet are below 360 GHz, whereas above approximately 10 km, both frequencies are above 370 GHz. These results indicate that no single set of triplets is optimal across all altitudes. The optimal bandwidth ranges from 30 GHz at higher altitudes to 10 GHz at lower altitudes.

At high altitudes (h $> 12$), the optimization algorithm selects frequencies that are more closely spaced to minimize the impact of large biases associated with the significantly dry conditions of this region. At these heights the differential reflectivity measurements are close to the radar measurement noise.

Figure 4b displays the precision of each optimal triplet shown in panel a. To provide context, the mean and range of model values are also displayed. Notably, the precision exceeds the expected retrieved values (i.e., precision $> 100\%$) at most heights.

However, precision can be enhanced by averaging successive retrieved values along the track, as discussed later.

Figure 4c displays the absolute percentage biases for each of the optimal triplets shown in panel a. Biases are well below 10% up to approximately 13 km. These precision and bias values represent the best achievable performance, given that the current instrument configuration can only accommodate one triplet at a time due to hardware constraints.





The best performance is found for humidity values between 0.1 and 1 gm$^{-3}$, which, in this simulation, occur at altitudes around 7–10 km. Within this range, the measurement precision at the single-pixel level is approximately 100%. The anticipated precision of DAR might initially appear large when compared to the uncertainty estimates of current upper tropospheric water vapor measurements. For instance, in the upper troposphere, the Atmospheric Infrared Sounder (AIRS) expected precision is around 20% (e.g., Susskind et al., 2003), the Microwave Limb Sounder (MLS) one is around 16% (Livesey et al., 2022), while the radiooccultation is about 20% (Kursinski et al., 1995). However, AIRS has a vertical resolution of around 4.3 km near the tropopause, with a measurement area, referred to as field of regard, of ∼40 km, while the MLS and radiooccultation measurements have a vertical resolution ranging from 1 to 2.3 km and an along track resolution of ∼200 km. In contrast, a single pixel DAR measurement provides a much finer resolution, with a 200 m vertically and only 400 m along track. Furthermore, as mentioned above, the DAR precision can be improved by a factor $\sqrt{N}$ averaging successive retrievals, as explored in the next section.

Although Figure 4 was discussed in terms of altitude, the water vapor burden has a greater impact on the optimal triplets (and their expected uncertainty) than altitude itself. Further, since water vapor is largely controlled by temperature, the choice of triplets is similarly influenced by it. Therefore, while the triplets presented in this figure are based on a tropical scenario, they can be applied to other latitude regions with comparable temperature (and thus, water vapor) conditions.

### 3.1 Precision and biases

GATE in-cloud retrievals using a single triplet across all heights are summarized in Figure 5a-e. In particular, we show retrievals for triplets optimized for clouds around 4, 5, 7, 9, and 11 km (as shown in Figure 4a). Overall, the mean retrieved values (across the entire LES domain) agree reasonably well with the conditionally sampled truth, that is, the retrievals match the mean model values when sampled at locations where the retrieval is feasible based on the radar frequencies used.

As an indication of the retrieval random error, the error bars display the standard deviation of the retrieved values at each height. As expected, the retrieved variability at higher altitudes increases when using triplets optimized for lower altitudes. For example, Figure 5a shows the retrieval results using the triplet optimized for 4 km, showing uncertainty well outside the model range above 9 km. In contrast, when the triplet optimized for 7 km is used, this higher uncertainty is mostly observed above 12 km (Figure 5c).

To further explore the impact of the selected triplet, Figure 6 compares the precision and biases of the same triplets shown in Figure 5. In comparison to the best achievable performance, the precision of a given triplet (Figure 6a) degrades significantly beyond about 500 meters from its optimized altitude. Outside of its target altitude, the precision remains relatively constant with height. On the other hand, biases (Figure 6b) increase roughly one kilometer away from the triplet target altitude. Triplets optimized for the lower altitudes have very large bias at higher altitudes, reaching levels up to 800%. Considering the degradation of precision and bias away from its optimal height, the triplet to be used when developing a DAR instrument must be carefully chosen based on the target height (as a proxy for temperature) and the types of clouds intended for study.

To illustrate how precision scales with along-track averaging, Figure 7a-e displays examples of the expected precision when averaging 10 or 100 km distances. To compute these examples, the GATE end-to-end retrievals were divided along-track at





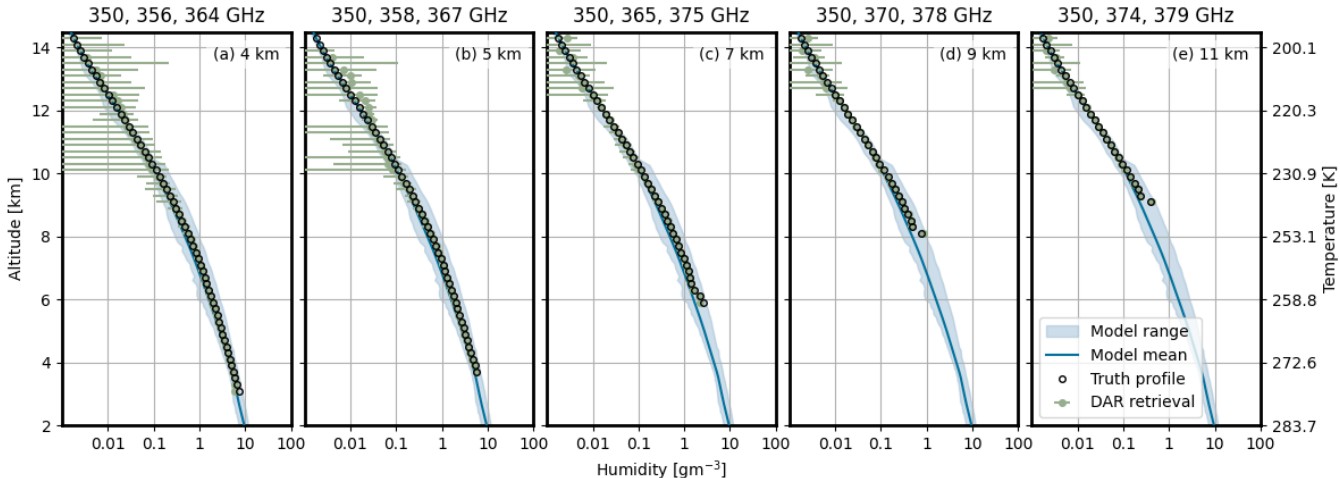

**Figure 5.** (a-e) GATE DAR retrievals (green dots) using different triplets optimized for 4, 5, 7, 9, and 11 km as indicated by the labels in each panel. Also shown are the LES domain mean model values (blue line), the range of model humidity at each height (light blue shaded area), and the conditionally sampled truth profile (black circles). The error bars depict the standard deviation of the retrieved values at each height, rather than indicating random measurement uncertainty.

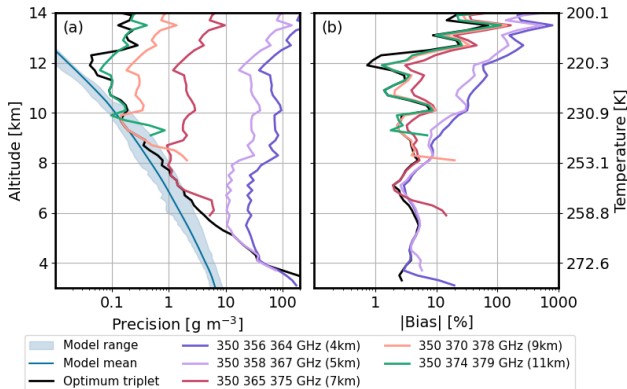

**Figure 6.** (a) Precision and (b) biases for selected triplets in comparison to the best achievable performance. The mean model values (blue line) and the range of model humidity (light blue shaded areas) are shown for reference in panel a.

the specified distances. Across each of these segments, the measured scene varies, offering different representations of the cloudiness a DAR instrument could encounter when measuring deep convective systems, such as the one represented in GATE.

The precision for each segment was computed using

$$\bar{\sigma}_e = \frac{\sqrt{\sum_i^N \sigma_{e_i}^2}}{N} \tag{6}$$





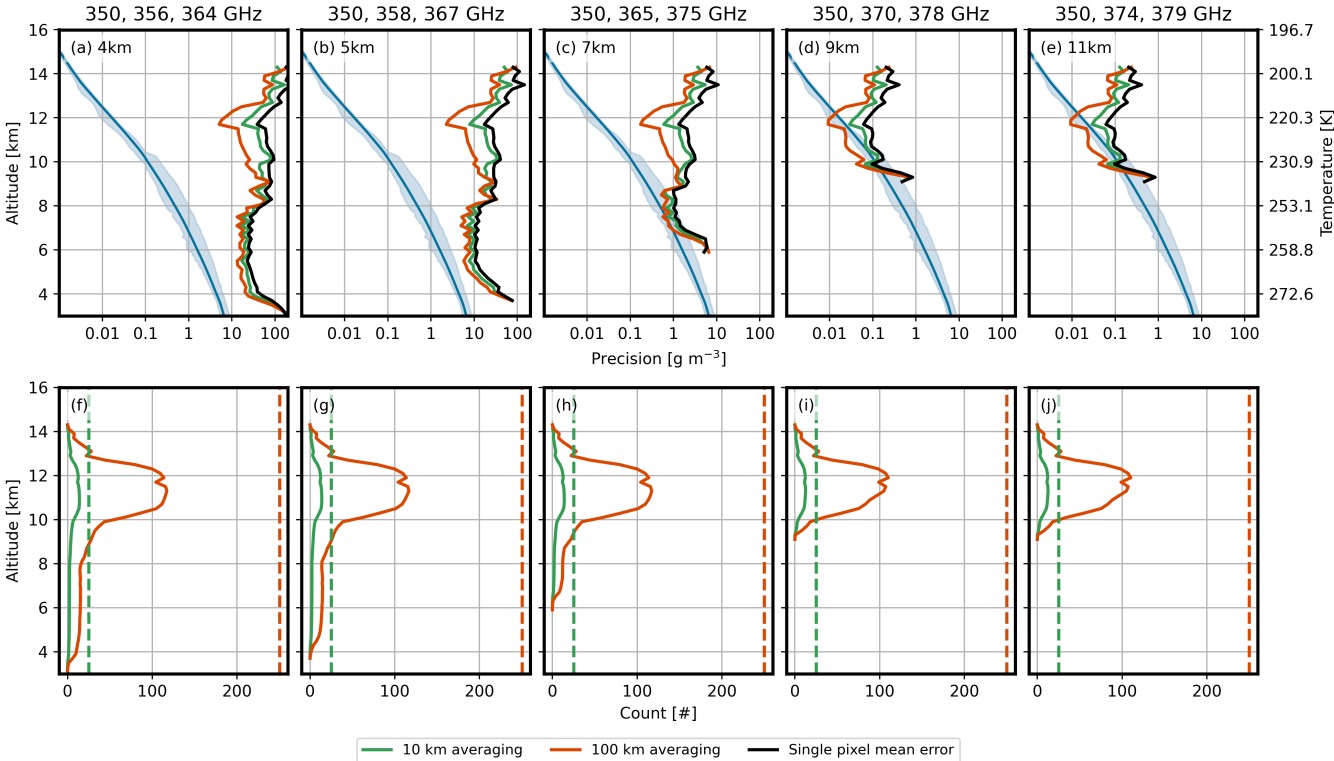

**Figure 7.** (a-e) Examples of retrieval precision using along-track averaging at different scales (e.g., 10 and 100 km) for different triplets. The mean model values (blue line) and the range of model humidity (light blue shaded areas) are shown for reference. (f-j) Average number of cloudy volumes measured within a specified along-track distance for different triplets. Dashed lines represent the maximum possible number of cloudy volumes within a given along track distance.

where $\sigma_{e_i}$ and $N$ are the precision errors and the number of measurements that contributed at each height. Figure 7a-e shows the mean precision (color lines) for these segments for different triplets, as well as, the mean precision for pixel-scale radar footprints across the entire domain (black line), provided as a reference.

Figure 7f-j also displays the mean number of cloudy volumes measured within specified along-track distances. The dashed lines indicate the maximum possible number of cloudy volumes for each distance, which is calculated by dividing the along-track distance by the footprint size. As shown, the mean number of cloudy volumes within a given along-track distance is significantly lower than the maximum possible, especially at lower altitudes. Put simply, it is unlikely that a cloud is present at every radar footprint for 10 km along-track distances and the likelihood diminishes even further over a 100 km span.

To better understand the possible precision improvements by across-track averaging, Figure 8 shows the precision when using the *optimal* triplets per height (shown in Figure 4). For the GATE environment, the greatest improvement of precision improvement when using along-track averaging occurs at around 11 km altitude, where most cloudy volumes are observed (see



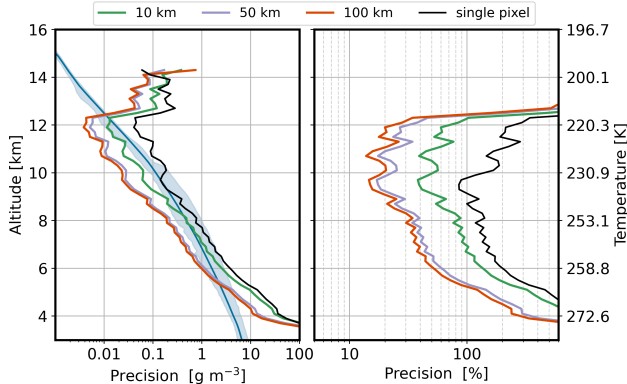

**Figure 8.** (a) Examples of retrieval precision using along-track averaging at different scales using the optimal triplets per height (shown in Figure 4). The mean model values (blue line) and the range of model humidity (light blue shaded areas) are shown for reference. (b) Retrieval precision as shown in panel a but in percent with respect to the mean model values.

Figure 7f-j). At this altitude, precision can improve significantly, from approximately 200% for a single radar pixel to around 20% when averaging over at least a 50 km across-track distance. Further, a DAR instrument with limited cross-track scanning
capability could enhance the number of cloud sampling opportunities. Essentially, a passive sensor could identify the cloudy path ahead of the radar measurements and provide this information to the DAR scanning antenna.

## 3.2 Yield

As discussed previously, the 350–380 GHz region explored here is well-suited for sampling mid and upper tropospheric clouds. To assess the global utility of the DAR technique to sample these clouds, we use CloudSat-driven simulations following Millán
et al. (2014, 2016) and Millán et al. (2020). In these simulations, instead of using LES fields, hydrometeor information was taken from CloudSat retrievals (Stephens et al., 2002; Austin and Stephens, 2001; Austin et al., 2009; Lebsock and L'Ecuyer, 2011), whereas temperature, pressure, and water vapor data were obtained from the European Centre for Medium-Range Weather Forecasts auxiliary (ECMWF-aux) products (Cronk and Partain, 2017).

The effectiveness of each of the triplets shown in Figure 5a-e largely depends on the atmospheric penetration of its most
attenuated frequency. Figure 9a-e displays the zonal yield of these frequencies to asses the global utility of such triplets. Most frequencies can achieve a global sampling yield of over 95% at the height for which their corresponding triplet was optimized. For example, 367 GHz can sample at 5 km globally with a yield exceeding 95%, while 375, 378, and 379 GHz can do so at 7, 9, and 11 km, respectively. The only exemption is the triplet optimized for 4 km. In the tropics, the yield for this frequency can drop to as low as 75%, due to the vast amounts of water vapor at these latitudes/altitude. Note that the yield improves
significantly with latitude, moving away from the equator. This result is intuitive as the amount of water vapor at a given altitude decreases at higher latitudes.



**Figure 9.** (a-e) CloudSat-driven yield zonal means (1-8 January 2007) for the most attenuated frequencies of the optimized triplets for 4, 5, 7, 9, and 11 km as indicated by the labels in each panel. White contours display the ice cloud occurrence in percentage (the levels shown are 1, 5, 10, 30%). (f-j) as panels a-e but using temperature as vertical axis.

Figure 9f-k displays the zonal yield for the same frequencies as in panels a-e but with respect to temperature instead of altitude. As shown, the yields remains nearly constant for a given temperature, regardless of latitude, showcasing that temperature, by controlling the water vapor burden, primarily determines the yield rather than altitude.

From Figure 9a-e it is evident that at any given latitude the yield drops sharply with altitude. Generally, within fewer than 2 km, the yield drops from over 95% to below 5%. In addition, Figure 9f-k demonstrate that the yield remains relatively stable with respect to temperature (which determines the water vapor burden). Therefore, careful consideration will be needed when developing a DAR instrument to select appropriate triplets based on the temperature range of the the types of clouds intended for study.





## 260   **4   Conclusions**

Current DAR instruments operate in the 158-174.8 GHz range due to international prohibitions on radar transmissions near the 183 GHz line center. These restrictions limit the accuracy of the retrieval making them impractical for cold and dry conditions with low water vapor content. In this study, we assessed the feasibility of using a spaceborne DAR operating near the 380 GHz unrestricted water vapor line to estimate water vapor inside mid and upper tropospheric clouds.

For this evaluation, we used a radar instrument simulator on LES fields to generate radar reflectivities, which were then used as measurements in end-to-end retrievals. In particular, we use LES simulations based on the GATE campaign, which is a tropical deep convective system in convective and radiative equilibrium. Further, we assumed an antenna diameter of 2 m and a transmitter with a peak output power of 100 W, a configuration that could be feasible in the near future.

End-to-end retrieval simulations were conducted using every possible triplet of radar tones between 350 and 380 GHz, spaced 270   1 GHz apart. Each retrieval enabled us to evaluate precision, yield, and associated biases. To identify the optimal triplet, we minimized the total error by summing the normalized precision and bias. Regardless of the height analyzed, the selected triplet always includes the most transparent available radar tone to anchor the retrieval. The spacing between the two other frequencies in the triplet ranges from 8 to 10 GHz. At lower altitudes, the algorithm chooses frequencies farther from the water vapor line to account for higher absorption. For example, below 4 km, both frequencies are under 360 GHz, while above 10 km, both are 275   above 370 GHz. Ultimately, the best triplet of radar tones depends on the water vapor burden (and its associated attenuation) at a given height. Since water vapor is largely controlled by temperature, the choice of triplets is similarly influenced by it.

In addition to identifying the optimal triplets, the end-to-end retrievals enabled us to examine the overall performance in terms of the precision, biases, and yield. In summary, single pixel precision (horizontal resolution $\simeq$ 400 m and vertical resolution = 200 m) are overall larger that 100%, degrading considerably away from the triplet's optimal height. However, 280   averaging along the track can significantly enhance the precision. For instance, at 11 km altitude, where most cloudy volumes (in the examined LES) occur, averaging over at least 50 km along-track can improve precision to around 20%. We note that this improvement may be less than expected since the actual number of cloudy volumes within a given along-track distance is often much lower than the theoretical maximum. Naturally, the extent of precision improvement from along-track averaging depends on the available retrievals, i.e., the number of cloudy volumes.

Along-track averaging cannot reduce biases; however, these biases are generally much smaller than precision errors, typically staying well below 10% when using the optimal triplet for a given height. While biases also degrade away from the triplet's optimal height, they do so to a lesser extent than precision.

Lastly, most radar tones examined here achieve a global sampling yield of over 95% at the height for which they were optimized. The only exception being the triplet designed for 4 km at the tropics, where the yield for this triplet can drop as 290   low as 75%, due to the vast amounts of water vapor in that region. From the yield assessment, it is evident that once the most attenuated frequency in the triplet begins to experience attenuation, the yield drops sharply, typically falling from over 95% to below 5% within just 2 km. Given the precision and bias degradation away from the optimal height —and the dramatic yield



drop— a triplet must be selected carefully when designing a DAR instrument, considering the target height (as proxy for the target temperature) and cloud types intended for study.

Overall, a spaceborne DAR instrument operating near the 380 GHz water vapor line shows potential to address shortcomings of current methods for measuring water vapor within upper tropospheric clouds, which are often inadequately represented in models.

*Code availability.* The radar simulator code used in this work will be made available upon request.

*Data availability.* The LES files are archived at https://doi.org/10.5281/zenodo.5544938 (Lebsock, 2021). All of the CloudSat data used in
this study are available from the CloudSat data processing center at https://www.cloudsat.cira.colostate.edu/data-products

*Author contributions.* LFM performed the analysis and wrote the manuscript. ML designed the project and provided guidance. MJK provided the LES fields. All the co-authors commented on and edited the manuscript.

*Competing interests.* The contact author has declared that none of the authors has any competing interests.

*Acknowledgements.* This research was carried out at the Jet Propulsion Laboratory, California Institute of Technology, under a contract with
the National Aeronautics and Space Administration (80NM0018D0004).



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
