# Peer review of "Retrieval simulations of a spaceborne differential absorption radar near the 380 GHz water vapor line"

_EGUsphere, 2025_

## Referee Comment (RC2)

Review for revised manuscript of " Retrieval simulations of a spaceborne differential absorption radar near the 380 GHz water vapor line" by Millán et al.

This paper evaluates the potential of using spaceborne differential absorption radar (DAR) operating near the 380 GHz water vapor absorption line to detect water vapor in the mid- and upper atmosphere, especially inside deep convective systems. The authors identify optimal radar frequency triplets that minimize precision errors and biases using large-eddy simulations (LES) and end-to-end retrieval experiments. The manuscript is well-structured and addresses a significant challenge in atmospheric remote sensing. The methodology is robust, combining LES-driven simulations with retrieval algorithms to assess DAR performance. However, there are some questions and comments that may need to be further considered and major revision is necessary.

**General Comments:**

Abstract: The vertical resolution in the manuscript is inconsistent with the vertical resolution in the table 2. What is the reason for this difference? It should be supplemented in the manuscript.

Introduction: Please provide a clearer explanation in the introduction section of the advantages of using the 380 GHz water vapor absorption line over the existing 183 GHz absorption line for retrieval, emphasizing the importance of this absorption line.

Table 1: The radar forward model description references Table 1, but the function expressions of the assumed particle size distribution are not explicitly listed in the table. Please add it in the revised manuscript.

**Specific Comments:**

Line 189: Correct "radioocculation" to "radio occultation".

Lines 216–220: The precision improvement via along-track averaging (Figure 7) is compelling, but the discussion lacks a quantitative comparison to existing instruments. Adding a paragraph contrasting DAR's precision after averaging with other methods would emphasize its novelty.

Line 258: Remove duplicate "the" in "the the types".

Line 285: The statement "biases are generally much smaller than precision errors" conflicts with Figure 6b, where biases for some triplets exceed 800%. Please check it.

---

## Author Response (AR1)

We thank the reviewers for their comments. Below are our responses in **blue**.

**Review by Davide Ori**

Thank you for the invitation to review this interesting study that explore the feasibility of a space-borne Differential Absorption Radar that operates in the sub-mm region near the 380 GHz water vapor absorption line. The authors listed a number of advantages of the suggested approach, primarily the fact that the investigated frequency band is outside of the internationally regulated microwave regions of the electromagnetic spectrum. Given the high extinction of the signal for these frequencies its applicability is however limited to the upper portion of the troposphere. The feasibility study is performed mostly upon model and reanalysis data, focusing on the optimization of the frequency selection in order to minimize the estimated bias and uncertainty of the retrieval. Retrieved Cloud-Sat products are used to assess the expected observation yield in a realistic scenario.

Overall, I find this study interesting and worth to be published. The text is well written and easy to follow, the figures and the data are of good quality and I wasn't able to find any major fallacy in the logic or the approaches employed. I would like to suggest a few improvements that I think can elevate the usefulness of the study.

Line 6 - Any differential measurement approach implies pairs of measurements, here it is abruptly introduced the fact that triplets are needed. Perhaps it is worth saying already in the abstract that the use of 3 frequencies are necessary to account for the confounding effect of differential scattering and extinction by hydrometeors and absorption by dry gases.

We modified the abstract, it now reads (new text in bold):

End-to-end retrieval simulations across the 350–380 GHz range were used to identify optimal radar frequency triplets, minimizing precision and biases, at each altitude. **While dual-frequency DAR systems can be susceptible to biases caused by range-dependent hydrometeor scattering, incorporating a third frequency allows for partial separation of water vapor extinction from the scattering and absorption effects of hydrometeors.** Each optimum triplet included the most transparent frequency available ...

Line 10 - here vertical resolution is 200m while in table 2 is going to be 50m. I guess the limitation comes from the vertical resolution of the model. But I do not think this discrepancy is well explained in the text. Perhaps it is worth mentioning it, and tell people that the result in the upper atmosphere are also sort of averaged over multiple range gates because of the limited vertical resolution of the model.

This discrepancy arises because there are 2 vertical resolutions discussed in the manuscript: the native radar range resolution and the retrieved water vapor vertical resolution (see section 2.3 for details).

To distinguish the two, we have updated table 2 to read Radar range resolution*  instead of just range resolution, and we added a note on the table that reads: *Note that the retrieved water vapor vertical resolution is 200 m (see section 2.3 for details).

We also updated the retrieval discussion. It previously read "To achieve this goal, we conducted end-to-end retrieval simulations assuming a retrieval step, R, of 200 m. The retrieval algorithm …"

 Now it states:

To achieve this goal, we conducted end-to-end retrieval simulations assuming a retrieval step, R, of 200 m. **This step effectively defines the vertical resolution of the retrieved water vapor estimates and should not be confused with the radar's range resolution.** The retrieval algorithm …

Line 71 - What exactly is used to represent ice and snow in the radar simulator? The text says hexagonal columns and dendrites, but the reference Leinonen and Szyrmer (2015) present a set of rimed and unrimed aggregates of dendrites and produce their scattering properties up to only 94 GHz. In Table 1 T-matrix and Mie scattering are mentioned which would imply spherical and spheroidal particles, is T-matrix used for ice and snow? Also, if Leinonen and Szyrmer (2015) shapes are used, are they consistent with the mass-size relation assumed in the WRF-1mom microphysics? Are the PSD references in Table 1 also the same in WRF or are they just assumed by the radar simulator?

Thanks for this comment. We did not provide sufficient referencing for our forward simulations. In short we use the model of Roy et al. (2021), which uses Leinonen and Szyrmer (2015) for some habits and T-matrix and Mie for others. Also the PSD and mass-size relationships are not consistent with the WRF simulation however the simulation only provides bulk quantities so there is no compelling reason to use the model assumptions over any other realistic assumption. We have modified the text as well as the table to provide more information, the text now reads (new text in bold):

Radar returns were simulated using the same radar forward model as discussed in Roy et al. (2021). In short, radar reflectivities were estimated using the time-dependent two-stream approximation (Hogan and Battaglia, 2008), assuming spherical particles for cloud, rain and graupel hydrometeors, hexagonal columns for ice, and dendrites for snow. **The water and ice dielectric properties were taken from Liebe et al. (1991) and Warren and Brandt (2008) respectively. The particle size distributions of all hydrometeor species**

**were represented using a modified gamma distribution. Specifics of these distributions can be found at Roy et al. (2021) appendix A. Although the microphysical parameterizations used in the radar simulator are not the same as those in the LES models, this discrepancy is unlikely to significantly affect the accuracy of the forward-simulated observations, as the LES models only supply bulk properties of such distributions.**

The table was updated to include the type of particle shape assumed for cloud, rain, ice, etc.

Section 2.3 - I would appreciate a cleaner description of the retrieval, at the moment it seems to me unnecessarily convoluted. I understand the logic, but it took me a while to grasp the discourse between lines 109 and 119. Perhaps I was off-put because I disagree with the statement on line 109-110: eq.2 does not allow to fit the terms on the righthand side by measuring gamma at various frequencies unless one makes assumptions on the functional forms of the such terms. Anyway, why not just mention the immunity to calibration errors and move on directly to the concept of linearizing the 3 confounding terms with respect to frequency?

We believe the current retrieval description strikes a good balance between simplicity and clarity, while still providing readers with sufficient detail to understand the key aspects of the methodology. However, our wording was not completely clear. We modified the opening sentence of that section, the updated text reads:

The principle behind DAR is based on the idea that by measuring the observed absorption coefficient, gamma(r1, r2, f)$ at different frequencies, enables the retrieval of the terms on the right-hand side of equation, given appropriate assumptions regarding the frequency and range dependence of particle single scattering properties. Specifically, the first term .....

Also, it is not immediately clear to me why the variability of the PSD is affecting the feasibility of a 2-frequency retrieval (line 117), the PSD is technically always the same (apart from beam-mismatches at various frequencies) and a third frequency would not help, the problem comes solely from the fact that all terms are frequency dependent so you need a way to transform the equation in to a system of equations with a limited number of unknowns.

The fundamental issue we are referencing is the variation in the PSD with range which causes the frequency dependence to change from one range to another. Most dramatically

this occurs at the melting level. We revised the text to further explain the origin of such biases.

The text now reads:

Assuming the last 3 parameters in equation 2 are a frequency-independent offsets, the humidity can be estimated directly using measurements of Ze at two frequencies (e.g., Roy et al., 2018). However, **in practice, this assumption breaks down in regions where the hydrometeor particle size distributions vary significantly with range, such as near cloud edges or in zones undergoing phase transitions. In such cases, the absorption coefficient, γ(r1, r2, f ), may be derived from reflectivities corresponding to different cloud regimes, such as a liquid cloud in r1 and an ice cloud in r2.** This range-dependent differential scattering of hydrometeors could be mistakenly attributed to water vapor attenuation, leading to humidity biases in the retrievals (e.g., Roy et al., 2020; Millán et al., 2024).

Line 151 - Do I understand correctly? You simulated all 31 frequencies between 350 and 380 GHz (extremes included) at 1 GHz step. Then you made experiments considering all possible triplets among these 31 frequencies, correct? Which if I am not mistaken it accounts for 8990 combinations. Perhaps you can also mention this number.

We added the following sentence (in bold):

We conducted end-to-end retrieval simulations for each frequency triplet in the 350 to 380 GHz region (with a 1 GHz spacing), similar to those shown in Figure 3, encompassing the entire GATE domain. **Specifically, we performed 4495 retrievals representing all possible combinations of three frequencies from the 31 available (i.e., n! / (n−r)!r! where n=31 and r=3).** At each height ...

Yield - I understand the definition, but I was wondering about a broader discussion on the benefit of this proposed observation platform in the context of existing satellite measurements. One of the great limitations of DAR techniques is the possibility of measuring only in clouds, this particular approach is even more limited to the upper portions of the atmosphere while DIAL can observe in clear-air. It might be interesting to make some quantitative assessment of the relative importance of the observational gap filled by this sub-mm DAR in the context of already established water-vapour retrieval techniques.

The spaceborne DAR precision and resolution was already put into perspective of the current technologies (MLS, radio occultation, and AIRS). To further emphasize the DAR capabilities, in the discussion of Figure 8, we added the text in bold (line **234** in the updated manuscript):

... At this altitude, precision can improve significantly, from approximately 200% for a single radar pixel to around 20% when averaging over at least a 50 km across-track distance. **This 50 km-averaged precision is comparable to that of current satellite instruments: 20% for AIRS, 16% for MLS, and 20% for radio occultation. Additionally, it offers significantly better vertical resolution (200 m versus 1–4.3 km) and better horizontal resolution compared to MLS and radio occultation (50 km versus 200 km), while being similar to the AIRS field of regard (50 km versus 40 km).** Further, a DAR instrument with limited cross-track scanning ...

Lastly we added at the end of the conclusion section:

Overall, a spaceborne DAR instrument operating near the 380 GHz water vapor line shows potential to address shortcomings of current methods for measuring water vapor within upper tropospheric clouds, which are often inadequately represented in models. **To further enhance observational capabilities, integrating DAR with DIAL on a single spaceborne platform would enable the retrieval of high-resolution water vapor profiles in both clear-sky and cloudy conditions, as demonstrated by Millán et al. (2024) through aircraft-based field campaign measurements. While DIAL signals are sensitive to backscatter from both aerosols and molecules, they are quickly attenuated in the presence of clouds. In contrast, DAR signals, due to their much longer wavelengths, primarily interact with larger particles, making them well-suited for probing cloudy and precipitating environments, as shown here.**

Note that a quantitative assessment of a spaceborne DIAL/DAR system is outside the scope of this study.

Trade offs - I found interesting observing the trade offs that one needs to make when selecting optimal frequency triplets. I was wondering if it would make sense to compensate for these limitations by introducing a fourth frequency? Not necessarily developing a full retrieval operating with frequency quads, but perhaps having a platform capable of operating multiple frequency triplets by selecting from a set of four available.

We added the following sentence in the manuscript (in bold): ... Considering the degradation of precision and bias away from its optimal height, the triplet to be used when developing a DAR instrument must be carefully chosen based on the target height (as a proxy for temperature) and the types of clouds intended for study. **We note that an instrument capable of frequency tuning or capable of transmission at more than three frequencies (even if not simultaneously) could potentially overcome this limitation, however such configurations are beyond the scope of this study.**

Precision - I understand this might be established jargon, but I find odd to use the term "precision" instead of "uncertainty". It just bothers me personally that this use of terms would imply that one would aim to reduce the value precision in order to get more precise results and the authors did a great job avoiding this problem by using terms such as

"degrade", or "enhance", and so on. Maybe just mention this oddity at line 136 where precision is defined as the spread of random retrieval errors. Alternatively ignore this point all-together, I will be fine.

The text now reads: The aim of this study is to explore various frequency combinations to identify triplets that not only reduce such biases but that also minimizes the retrieved humidity precision (i.e., those associated with the reflectivity random errors**, commonly referred to as the retrieval uncertainty).** To achieve this goal, …

**Reviewer 2:**

Review for revised manuscript of " Retrieval simulations of a spaceborne differential absorption radar near the 380 GHz water vapor line" by Millán et al.

This paper evaluates the potential of using spaceborne differential absorption radar (DAR) operating near the 380 GHz water vapor absorption line to detect water vapor in the mid- and upper atmosphere, especially inside deep convective systems. The authors identify optimal radar frequency triplets that minimize precision errors and biases using large-eddy simulations (LES) and end-to-end retrieval experiments. The manuscript is well-structured and addresses a significant challenge in atmospheric remote sensing. The methodology is robust, combining LES-driven simulations with retrieval algorithms to assess DAR performance. However, there are some questions and comments that may need to be further considered and major revision is necessary.

General Comments:

Abstract: The vertical resolution in the manuscript is inconsistent with the vertical resolution in the table 2. What is the reason for this difference? It should be supplemented in the manuscript.

This discrepancy arises because there are 2 vertical resolutions discussed in the manuscript: the native radar range resolution and the retrieved water vapor vertical resolution (see section 2.3 for details).

To distinguish the two, we have updated table 2 to read Radar range resolution* instead of just range resolution, and we added a note on the table that reads: *Note that the retrieved water vapor vertical resolution is 200 m (see section 2.3 for details).

We also updated the retrieval discussion. It previously read "To achieve this goal, we conducted end-to-end retrieval simulations assuming a retrieval step, R, of 200 m. The retrieval algorithm …"

 Now it states:

To achieve this goal, we conducted end-to-end retrieval simulations assuming a retrieval step, R, of 200 m. **This step effectively defines the vertical resolution of the retrieved water vapor estimates and should not be confused with the radar's range resolution.** The retrieval algorithm …

Introduction: Please provide a clearer explanation in the introduction section of the advantages of using the 380 GHz water vapor absorption line over the existing 183 GHz absorption line for retrieval, emphasizing the importance of this absorption line.

It is unclear what specific importance the reviewer is referring to, as the current introduction already notes that the 380 GHz line is unregulated, in contrast to the 183 GHz line, for which transmission is prohibited. We even included a figure to highlight this advantage of the 380 GHz line over the 183 GHz line.

We also mention that the 380 spectral region  "could extend the DAR technique to mid and upper tropospheric clouds, including those associated with convection and stratiform anvils. These lines are strongly absorbing, maximizing DAR sensitivity in these regions where vapor abundance (and thus absorption) is low."

Table 1: The radar forward model description references Table 1, but the function expressions of the assumed particle size distribution are not explicitly listed in the table. Please add it in the revised manuscript.

After careful consideration, we decided not to include explicitly the PSDs expressions, however, we refer the reader to Appendix A of Roy et al. (2021), here all relevant information is given.

The text now reads (new text in bold):

Radar returns were simulated using the same radar forward model as discussed in Roy et al. (2021). In short, radar reflectivities were estimated using the time-dependent two-stream approximation (Hogan and Battaglia, 2008), assuming spherical particles for cloud,

rain and graupel hydrometeors, hexagonal columns for ice, and dendrites for snow. **The water and ice dielectric properties were taken from Liebe et al. (1991) and Warren and Brandt (2008) respectively. The particle size distributions of all hydrometeor species were represented using a modified gamma distribution. Specifics of these distributions can be found at Roy et al. (2021) appendix A. Although the microphysical parameterizations used in the radar simulator are not the same as those in the LES models, this discrepancy is unlikely to significantly affect the accuracy of the forward-simulated observations, as the LES models only supply bulk properties of such distributions.**

Specific Comments:

Line 189: Correct "radioocculation" to "radio occultation". This typo was corrected through-out the manuscript

Lines 216–220: The precision improvement via along-track averaging (Figure 7) is compelling, but the discussion lacks a quantitative comparison to existing instruments. Adding a paragraph contrasting DAR's precision after averaging with other methods would emphasize its novelty.

In the discussion of Figure 8 we added the text in bold (line **234** in the updated manuscript):

… At this altitude, precision can improve significantly, from approximately 200% for a single radar pixel to around 20% when averaging over at least a 50 km across-track distance. **This 50 km-averaged precision is comparable to that of current satellite instruments: 20% for AIRS, 16% for MLS, and 20% for radio occultation. Additionally, it offers significantly better vertical resolution (200 m versus 1–4.3 km) and better horizontal resolution compared to MLS and radio occultation (50 km versus 200 km), while being similar to the AIRS field of regard (50 km versus 40 km).** Further, a DAR instrument with limited cross-track scanning …

Note that those satellite instruments (and their precision and vertical and horizontal resolution) were previously introduced.

Line 258: Remove duplicate "the" in "the the types". Done

Line 285: The statement "biases are generally much smaller than precision errors" conflicts with Figure 6b, where biases for some triplets exceed 800%. Please check it.

The reviewer is correct to highlight this statement; it is indeed a nuanced point that depends on the clause "when using the optimal triplet for a given height". This clause refers specifically to examining Figure 6b only at the altitude for which each triplet is optimized (as indicated in the legend in brackets for each triplet). To clarify this, we have rewritten the sentence to showcase that clause up front. The sentence now reads:

Along-track averaging cannot reduce biases; however, *when using the optimal triplet for a given height*, these biases are generally much smaller than precision errors, typically staying well below 10%.